# Pre-Existing Interstitial Lung Abnormalities Are Independent Risk Factors for Interstitial Lung Disease during Durvalumab Treatment after Chemoradiotherapy in Patients with Locally Advanced Non-Small-Cell Lung Cancer

**DOI:** 10.3390/cancers14246236

**Published:** 2022-12-17

**Authors:** Wakako Daido, Takeshi Masuda, Nobuki Imano, Naoko Matsumoto, Kosuke Hamai, Yasuo Iwamoto, Yusuke Takayama, Sayaka Ueno, Masahiko Sumii, Hiroyasu Shoda, Nobuhisa Ishikawa, Masahiro Yamasaki, Yoshifumi Nishimura, Shigeo Kawase, Naoki Shiota, Yoshikazu Awaya, Tomoko Suzuki, Soichi Kitaguchi, Kazunori Fujitaka, Yasushi Nagata, Noboru Hattori

**Affiliations:** 1Department of Respiratory Medicine, Hiroshima University Hospital, Hiroshima 734-8551, Japan; 2Department of Radiation Oncology, Hiroshima University Hospital, Hiroshima 734-8551, Japan; 3Department of Respiratory Internal Medicine, Hiroshima Red Cross Hospital & Atomic-Bomb Survivors Hospital, Hiroshima 730-8619, Japan; 4Department of Respiratory Medicine, Hiroshima Prefectural Hospital, Hiroshima 734-8530, Japan; 5Department of Medical Oncology, Hiroshima City Hiroshima Citizens Hospital, Hiroshima 730-8518, Japan; 6Department of Respiratory Internal Medicine, Hiroshima City Hiroshima Citizens Hospital, Hiroshima 730-8518, Japan; 7Department of Respiratory Medicine, National Hospital Organization Higashihiroshima Medical Center, Hiroshima 739-0041, Japan; 8Department of Respiratory Medicine, Kure Kyosai Hospital, Hiroshima 737-8505, Japan; 9Department of Respiratory Disease, Chugoku Rousai General Hospital, Hiroshima 737-0193, Japan; 10Department of Respiratory Medicine, Miyoshi Central Hospital, Hiroshima 728-8502, Japan; 11Department of Respiratory Medicine, JA Onomichi General Hospital, Hiroshima 722-8508, Japan; 12Department of Respiratory Internal Medicine, Hiroshima City North Medical Center Asa Citizens Hospital, Hiroshima 731-0232, Japan

**Keywords:** chemoradiotherapy, durvalumab, interstitial lung abnormalities, interstitial lung disease

## Abstract

**Simple Summary:**

Interstitial lung disease (ILD) is a life-threatening toxicity caused by chemoradiotherapy and durvalumab; however, the risk factors for ILD during durvalumab treatment after chemoradiotherapy have not been established in non-small-cell lung cancer patients. We examined whether interstitial lung abnormalities (ILAs) could be risk factors for ≥grade-two ILD during durvalumab treatment. The prevalence of ILAs in 148 patients before durvalumab treatment was 37.8%, and the multivariate analysis revealed ILAs as independent risk factors for ≥grade-two ILD. Attention should be paid to the development of ILD during durvalumab treatment in patients with ILAs.

**Abstract:**

Introduction/Background: Chemoradiotherapy (CRT) followed by durvalumab, an immune checkpoint inhibitor, is the standard treatment for locally advanced non-small-cell lung cancer (NSCLC). Interstitial lung disease (ILD) is a life-threatening toxicity caused by these treatments; however, risk factors for the ILD have not yet been established. Interstitial lung abnormalities (ILAs) are computed tomography (CT) findings which manifest as minor interstitial shadows. We aimed to investigate whether ILAs could be risk factors for grade-two or higher ILD during durvalumab therapy. Patients and Methods: Patients with NSCLC who received durvalumab after CRT from July 2018 to June 2021 were retrospectively enrolled. We obtained patient characteristics, laboratory data, radiotherapeutic parameters, and chest CT findings before durvalumab therapy. Results: A total of 148 patients were enrolled. The prevalence of ILAs before durvalumab treatment was 37.8%. Among 148 patients, 63.5% developed ILD during durvalumab therapy. The proportion of patients with grade-two or higher ILD was 33.8%. The univariate logistic regression analysis revealed that older age, high dose-volume histogram parameters, and the presence of ILAs were significant risk factors for grade-two or higher ILD. The multivariate analysis showed that ILAs were independent risk factors for grade-two or higher ILD (odds ratio, 3.70; 95% confidence interval, 1.69–7.72; *p* < 0.001). Conclusions: We showed that pre-existing ILAs are risk factors for ILD during durvalumab treatment after CRT. We should pay attention to the development of grade-two or higher ILD during durvalumab treatment in patients with ILAs.

## 1. Introduction

Lung cancer is a major malignant tumor, with a high incidence and mortality [1]. It is categorized into two major histological subtypes, non-small-cell lung cancer (NSCLC) and small-cell lung cancer (SCLC), which account for approximately 80% and 13%, respectively [2,3]. Recently, the decline in mortality for lung cancer, especially NSCLC, has accelerated [1,2] owing to therapeutic advances, including immunotherapy (i.e., inhibitors of programmed cell death protein-1 (PD-1)/programmed death ligand-1 (PD-L1)) and molecular targeted therapy.

Durvalumab, an anti-PD-L1 antibody, is an immune checkpoint inhibitor (ICI). The PACIFIC study, a randomized phase III trial, compared durvalumab and a placebo as maintenance therapy for locally advanced unresectable NSCLC patients after curative chemoradiotherapy (CRT). The primary endpoints were progression-free survival and overall survival, which were better in the durvalumab group than in the placebo group [4,5,6,7]. This favorable result makes durvalumab a standard therapy for locally advanced unresectable NSCLC after curative CRT [5,8].

Interstitial lung disease (ILD) is a major life-threatening adverse event caused by ICIs, chemotherapy, and radiotherapy. A systematic review and meta-analysis showed that the incidence of ICI-induced ILD was relatively higher in NSCLC than in other types of cancers [9,10], and 10% of cases were mortal [11]. The PACIFIC study revealed that the incidence of ILD of any grade in the durvalumab group was 33.9% [4]. Additionally, real-world data showed a higher incidence of ILD (approximately 60–80%) related to the PACIFIC regimen, especially durvalumab [12,13,14,15,16,17,18]. The possible risk factors were radiotherapeutic parameters, such as the volume of the lung parenchyma that received 20 Gy (V20) or the mean lung dose (MLD) [12,13,17]. However, these factors were not significant in other studies [15,16,18]. Therefore, it is necessary to identify the risk factors for ILD during durvalumab treatment.

Interstitial lung abnormalities (ILAs) are defined as subtle or mild parenchymal abnormalities identified in more than 5% of any lung zone on computed tomography (CT) scans in patients not previously clinically suspected of ILD [19,20]. Although ILAs are subclinical findings, their clinical importance is increasing [21]. We have reported that ILAs were risk factors for ICI-induced ILD in patients with NSCLC [22] and even in patients with cancers other than lung cancer [23]. Thus, we hypothesized that ILAs were also risk factors for ILD during durvalumab therapy.

Here, we performed a multicenter retrospective study to investigate whether clinicopathological characteristics and ILAs could be risk factors for ILD during durvalumab treatment.

## 2. Materials and Methods

### 2.1. Study Design and Participants

Consecutive patients with unresectable NSCLC treated with CRT followed by durvalumab between July 2018 and June 2021 were retrospectively enrolled from 10 institutions. The patients who had received systemic corticosteroids before durvalumab treatment were excluded. Information on patient characteristics before durvalumab treatment, including chest CT findings and laboratory data, was obtained. The opt-out method was used in this study to obtain patient consent. This study was approved by the institutional review board of Hiroshima University (No. E-1590) and the ethics committee of each participating institution and was conducted in accordance with the ethical standards established by the Helsinki Declaration of 1975. This study followed the STROBE Reporting Guidelines for Observational Studies.

### 2.2. Assessment of CT Findings

We investigated the presence of interstitial pneumonia, pre-existing ILAs, and other abnormal findings on CT images taken at the end-inspiration phase in the supine position from the end of CRT before the start of durvalumab treatment. The types of ILAs (Appendix A) were determined and categorized as follows: ground glass attenuation (GGA), reticulation, honeycombing, centrilobular nodularity, traction bronchiectasis, nonemphysematous cysts on CT images, and non-dependent abnormalities affecting more than 5% of any lung zone (upper, middle, and lower lung zones were demarcated by the levels of the inferior aortic arch and right inferior pulmonary vein) [20]. The CT images were evaluated by two pulmonologists who did not have information on the characteristics of each patient. First, Readers 1 and 2 independently evaluated all the CT images. CT images with discordant results by the two readers were evaluated by Reader 3, and major opinion was considered the final evaluation of those images.

### 2.3. Diagnosis of ILD during Durvalumab Treatment

The diagnosis of ILD during durvalumab treatment was defined as follows: (1) new abnormal shadows found on chest CT; (2) exclusion of bacterial pneumonia (that did not improve even after administration of antibiotic drugs, or absence of bacterial pathogens detected by sputum culture); (3) exclusion of heart failure (by laboratory data and/or transthoracic echocardiography findings; and (4) exclusion of tumor progression (using laboratory data and version 1.1 of the Response Evaluation Criteria in Solid Tumors). ILD during durvalumab treatment in this study includes both radiation pneumonitis and ICI-induced ILD. ILD was classified according to pneumonitis based on the pneumonitis described in Common Terminology Criteria for Adverse Events (CTCAE) v5.0.

### 2.4. Statistical Analysis

The results are expressed as median (range). Comparisons between the two groups were performed by using Pearson’s chi-square or Wilcoxon tests. Univariate and multivariate logistic regression analyses were used to identify risk factors for grade 2 or higher (grade ≥ 2) ILDs. Factors with a *p*-value less than 0.05 in the univariate analyses were selected for inclusion in multivariable analysis. In addition, V20 in the radiotherapeutic parameters was selected for the multivariate analysis since V20 has been reported as a predictive dose–volume parameter for radiation pneumonitis in several studies [12,13,17,24,25,26,27,28]. To determine which GGA or reticulation in the ILAs was an independent risk factor for ILD, we generated Models A and B in the multivariate analysis. The optimal cutoff values of continuous variables such as the white blood cell (WBC) count, V5, V20, and MLD were estimated by using receiver operating characteristic curve analysis. The time of onset of ILD was analyzed by using the Kaplan–Meier method. All reported *p*-values were two-sided. Statistical significance was set at *p* < 0.05. All statistical analyses were performed by using JMP Pro 16 software (SAS Institute Inc., Cary, NC, USA).

## 3. Results

### 3.1. Patient Characteristics

A total of 153 patients were enrolled in this study. A flowchart of the patient selection process is shown in Appendix A. Ultimately, 148 patients were included in the analysis. The clinical characteristics and laboratory findings of the patients, as well as information regarding CRT and CT findings before durvalumab treatment are presented in Table 1. Male patients (71.6%), patients with Eastern Cooperative Oncology Group performance status (ECOG-PS) of 0 or 1 (98.0%), and current and former smokers (84.5%) were predominant. The majority of patients (90.5%) had stage III disease.

### 3.2. Incidence and Severity of ILD during Durvalumab Treatment

The occurrence and severity of ILD during durvalumab therapy are shown in Table 2. Among the 94 patients in whom ILD occurred after administration of durvalumab, 50 patients (33.8%) developed grade ≥2 ILD, 54 patients (36.5%) were led to discontinuation of durvalumab treatment, and 40 (27.0%) were administered systemic corticosteroids. ILD occurred within 16 weeks in approximately 90% of patients who developed ILD (Figure 1). We also present a case of grade-three ILD in a patient with ILAs (Appendix A).

### 3.3. Comparison of Characteristics between Patients with and without Grade ≥2 ILD

When comparing the baseline characteristics of each group (Table 3), patients with grade ≥2 ILD were significantly older than those without. Radiotherapy parameters such as V5, V20, and MLD were significantly higher in the patient group with grade ≥2 ILD than in that without. The proportion of patients with ILAs on CT was significantly higher in the group with grade ≥2 ILD than in that without (60.0% vs. 26.5%, respectively, *p* < 0.001). The proportions of patients with GGA and reticulation were significantly higher in the study group with grade ≥2 ILD than in that without (56.0% vs. 13.3%, *p* < 0.001; 32.0% vs. 14.3%, *p* = 0.011; respectively).

### 3.4. Logistic Regression Analysis of the Risk Factors for Grade ≥2 ILD

In the univariate logistic regression analysis, age ≥65 years (odds ratio (OR) 4.15, 95% confidence interval (CI): 2.02–8.55, *p* = 0.001), higher V5 (OR 3.68, 95% CI: 1.78–7.63, *p* < 0.001), higher V20 (OR 2.81, 95% CI: 1.39–5.68, *p* = 0.005), higher MLD (OR 3.66, 95% CI: 1.78–7.54, *p* < 0.001), and pre-existing ILAs (OR 4.15, 95% CI: 2.02–8.55, *p* = 0.001) were significant risk factors for grade ≥2 ILD (Table 4). We used two multivariate logistic regression analysis models (Table 5). Model A showed that ILAs (OR 3.70, 95% CI: 1.69–7.72, *p* = 0.001) and V20 (OR 2.64, 95% CI: 1.24–5.62, *p* = 0.012) were independent risk factors for grade ≥2 ILD. In Model B, GGA (OR 6.71, 95% CI: 2.80–16.08, *p* < 0.001) in the ILAs was an independent risk factor.

### 3.5. Risk Factors for Early Onset Grade ≥2 ILD

To identify risk factors for early onset (within 8 weeks after administration of durvalumab) grade ≥2 ILD, we performed a logistic regression analysis, using factors including existence of ILAs and radiation parameters (V5, V20, and MLD). In the univariate logistic regression analysis, pre-existing ILAs (OR, 6.25; 95% CI, 2.29–17.03; *p* < 0.001), higher V5 (OR, 4.58; 95% CI, 1.60–13.11; *p* = 0.005), and higher V20 (OR, 3.32; 95% CI, 1.27–8.64; *p* = 0.014) were significant risk factors for early onset grade ≥ 2 ILD (Table 6). In multivariate logistic progression analysis models (Table 7), pre-existing ILAs (OR, 6.19; 95% CI, 2.22–17.24; *p* < 0.001) and higher V20 (OR, 3.27; 95% CI, 1.20–8.92; *p* = 0.021) were independent risk factors for early onset grade ≥2 ILD. As shown in Figure 2, the incidence of early onset grade ≥ 2 ILD in patients with ILAs and a higher V20 was significantly higher than for those with ILAs or a higher V20 (42.9% vs. 13.4%, *p* = 0.026) and those without ILAs and with a lower V20 (42.9% vs. 3.8%, *p* < 0.001). The Kaplan–Meier analysis showed that the time to onset of grade ≥2 ILD was significantly shorter in the group with ILAs and a higher V20 compared to the group with ILAs or a higher V20, log-rank test *p* < 0.001; and in the group without ILAs and with lower V20, log-rank test *p* < 0.001 (Figure 3).

## 4. Discussion

In this study, the multivariate logistic regression analysis revealed that V20 and ILAs, especially GGA, were independent risk factors for grade ≥2 ILD. To the best of our knowledge, this is the first study to show that pre-existing ILAs are independent risk factors for grade ≥2 ILD during durvalumab treatment. Previous studies showed that the prevalence of ILAs was 4–9% in smokers and 2–7% in non-smokers [29,30,31,32]. Reportedly, ILAs progress to advanced fibrosis and are associated with decreased lung capacities and exercise capacities and high rates of mortality due to respiratory disease and all-cause mortality [19,21,29,33,34]. We have already reported that ILAs are risk factors for ICI-induced ILD in NSCLC patients and even in patients with cancers other than lung cancer [22,23,35]. Based on the results of this and previous studies, attention should be paid to the development of grade ≥2 ILD during durvalumab treatment in patients with ILAs.

The finding that ILAs, especially GGA, but not reticulation or honeycombing, is an independent risk factor for grade ≥2 ILD during durvalumab treatment is consistent with that of previous studies [22,23]. Durvalumab exerts antitumor effects by blocking immune tolerance and upregulating lymphocyte activity. GGA reflects the infiltration of lymphocytes into the interstitium of the lungs. In contrast, reticulation reflects slight fibrosis and lymphocytic inflammation, and honeycombing reflects the destruction of the normal alveolar structure [36,37]. These data explain why the presence of GGA in ILAs was associated with ICI during durvalumab treatment in this study. Based on these observations, we should consider ILD development in patients with GGA during durvalumab treatment.

Furthermore, we showed that V20, in addition to ILAs, was an independent risk factor for grade ≥2 ILD. Older age, high V20, high KL-6 level, and smoking history were reported as risk factors for radiation pneumonitis before the approval of durvalumab for CRT [23,24,25,26,27]. Recent studies have shown that a higher V20 or MLD is a risk factor for ILD in patients treated with durvalumab after CRT [12,13,17]. Our study also revealed that V20 was an independent risk factor for ILD. Thus, it is important to focus on V20 in addition to ILAs for predicting severe ILD during durvalumab treatment.

The incidence of ILD (any grade: 63.5%) in our study was higher than that in the PACIFIC study (any grade: 33.9%) [4]. The other real-world data and meta-analysis [12,13,14,15,16,17,18] also showed a higher prevalence of ILD than that in the PACIFIC study. This difference can be explained by the following three factors. First, ethnic differences exist between the PACIFIC study and real-world data. The pooled analysis in a real-world setting showed that the occurrence of ILD during durvalumab treatment after CRT for stage III NSCLC in Asian patients was significantly higher than that in Western patients [38]. The second difference was in the number of elderly patients. The present and previous studies [13,14,18] included a larger number of elderly patients than did the PACIFIC study; this difference might have influenced the frequency of ILD. Finally, MLD and V20 may also influence the prevalence of ILD. In the PACIFIC study, the MLD or V20 was limited to less than 35%. However, several patients in the present and previous studies received radiotherapy that exceeded these limits [12,15,16,18].

## 5. Limitations

First, this was a retrospective study. Further prospective studies are required to validate our findings. Second, we did not diagnose ILD pathologically and exclude the influence of concomitant drugs and environment completely. Pathological observations can be valuable in discussing the relationship between CT and pathological findings.

## 6. Conclusions

Our study showed that pre-existing ILAs, especially GGA, were significant risk factors for grade ≥2 ILD during durvalumab treatment. This observation is consistent with previous reports from our laboratory. Therefore, significant attention should be paid to the CT findings before durvalumab therapy. We believe that the observations from our study will be useful for the management of durvalumab treatment.

## Figures and Tables

**Figure 1 cancers-14-06236-f001:**
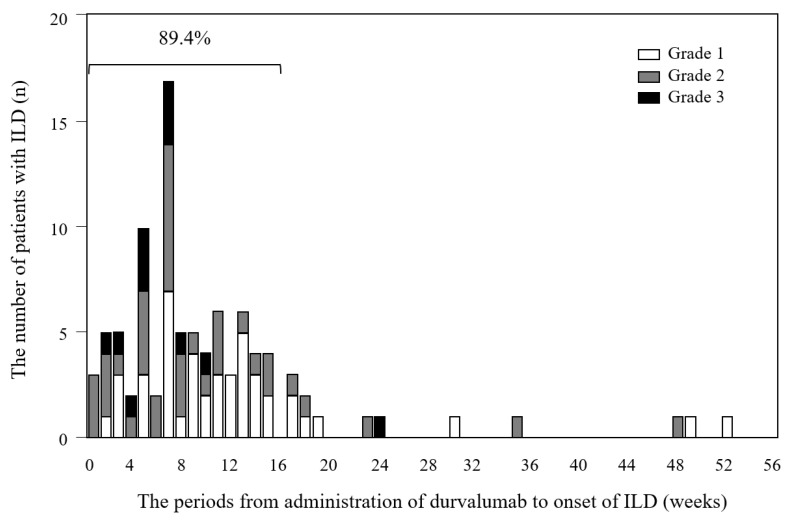
Onset, duration, and different grades of ILD during durvalumab treatment. ILD, interstitial lung disease.

**Figure 2 cancers-14-06236-f002:**
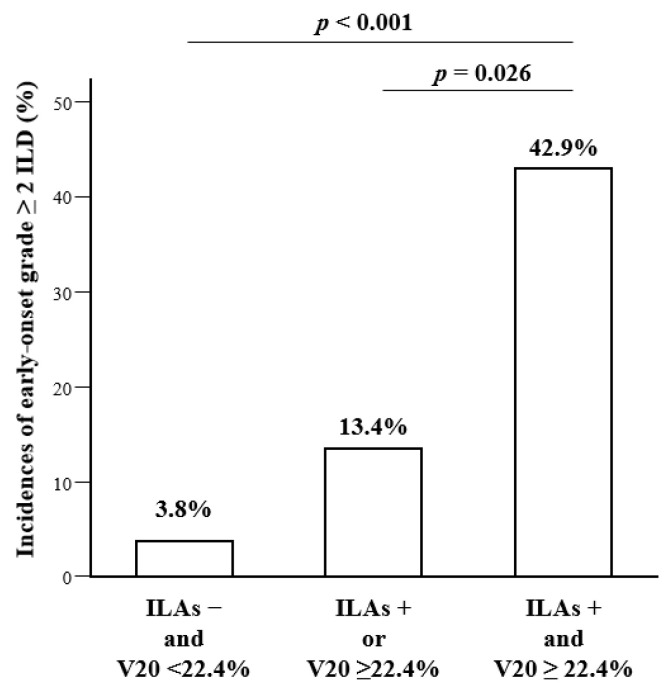
Incidence of early onset grade ≥2 ILD. Incidence of early onset grade ≥2 ILD in patients with ILAs and higher V20 was significantly higher than those with ILAs or higher V20 (42.9% vs. 13.4%, *p* = 0.0026) and those without ILAs and with lower V20 (42.9% vs. 3.8%, *p* < 0.001). ILAs, interstitial lung abnormalities; V20, volume of lung parenchyma that received ≥20 Gy; ILD, interstitial lung disease.

**Figure 3 cancers-14-06236-f003:**
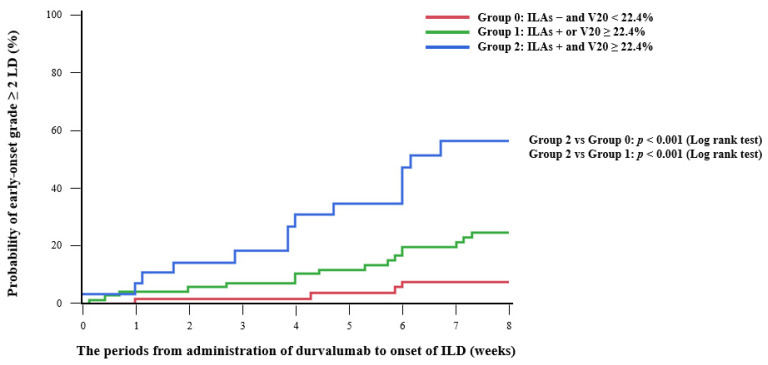
Kaplan–Meier analysis of time to onset of grade ≥2 ILD between patients with ILAs and higher V20, with ILAs or higher V20, and without ILAs and with lower V20. The time to onset of early onset grade ≥2 ILD was significantly shorter in the group with ILAs and higher V20 compared to the group with ILAs or higher V20, log-rank test *p* < 0.001; the group without ILAs and with lower V20, log-rank test *p* < 0.001. ILAs, interstitial lung abnormalities; V20, volume of lung parenchyma that received ≥20 Gy; ILD, interstitial lung disease.

**Table 1 cancers-14-06236-t001:** Baseline patient characteristics and laboratory findings.

Patient Characteristics	Patients, No. (%) (n = 148)
Age, years median (range)	71 (43–86)
Sex Male/female	106 (71.6)/42 (28.4)
BMI median (range)	21.2 (13.4–31.2)
ECOG PS 0/1/2	111 (75.0)/34 (23.0)/3 (2.0)
Smoking status Non-smokers/former or current smokers	23 (15.5)/125 (84.5)
Brinkman Index median (range)	857.5 (0–3840)
Autoimmune disease +/−	7 (4.7)/141 (95.3)
Clinical stage II/III/IV/postoperative recurrence	12 (8.1)/134 (90.5)/1 (0.7)/1 (0.7)
Histological diagnosis Adenocarcinoma/squamous cell carcinoma/others	77 (52.0)/64 (43.2)/7 (4.7)
Driver mutation EGFR/ALK/-/not tested	14 (9.5)/1 (0.7)/85 (57.4)/48 (32.4)
PD-L1 TPS <1%/1–49%/≥50%/not tested	29 (19.6)/35 (23.6)/41 (27.7)/43 (29.1)
Laboratory findings	Median (range) (n = 148)
WBC count (/μL)	3830 (1300–8880)
Lymphocyte count (/μL)	730 (180–5000)
LDH level (IU/L)	192 (126–413)
CRP level (mg/dL)	0.485 (0.016–13.58)
KL-6 level (U/mL)	295 (116–2221)
SpO_2_ level (%)	98 (92–100)
Information of CRT	Patients, No. (%) (n = 148)
Concurrent chemotherapy	
Weekly carboplatin + paclitaxel	87 (58.8)
Cisplatin + S-1	18 (12.2)
Cisplatin + vinorelbine	34 (23.0)
Daily carboplatin	5 (3.4)
Carboplatin + paclitaxel	1 (0.7)
Cisplatin + docetaxel	3 (2.0)
Total irradiation dose, Gy median (range)	65 (40–70)
V5, % median (range)	42.3 (10.7–83.2)
V20, % median (range)	21.6 (2.9–40.7)
Mean lung dose, Gy median (range)	12.1 (2.8–19.2)
Radiation technique, (%) 3D/IMRT/3D+IMRT	84 (56.8)/53 (35.8)/11 (7.4)
Best tumor response, (%) CR/PR/SD/PD/not evaluated	6 (4.1)/102 (68.9)/38 (25.7)/1 (0.7)/1 (0.7)
Interval between CRT and durvalumab, days median (range)	17 (1–54)
CT findings	Patients, No. (%) (n = 148)
IP +/−	2 (1.4)/146 (98.6)
Emphysema +/−	82 (55.4)/66 (44.6)
ILAs +/−	56 (37.8)/92 (62.2)
Type of ILAs	
GGA +/−	41 (27.7)/107 (72.3)
Reticulation +/−	30 (20.1)/118 (80.8)
Honeycombing +/−	1 (0.7)/147 (99.3)
Other +/−	6 (4.1)/142 (95.9)

BMI, body mass index; ECOG PS, Eastern Cooperative Oncology Group performance status; EGFR, epidermal growth factor receptor; ALK, anaplastic lymphoma kinase; PD-L1, programmed death-ligand 1; TPS, tumor proportion score; WBC, white blood cell; LDH, lactate dehydrogenase; CRP, C-reactive protein; CRT, chemoradiotherapy; V5, the volume of lung parenchyma that received ≥5 Gy; V20, volume of lung parenchyma that received ≥20 Gy; IMRT, intensity modulated radiation therapy; CR, complete response; PR, partial response; SD, stable disease; PD, progressive disease; CT, computed tomography; IP, interstitial pneumonitis; ILAs, interstitial lung abnormalities; GGA, ground glass attenuation.

**Table 2 cancers-14-06236-t002:** Occurrence and severity of ILD during durvalumab therapy.

Characteristics	Patients, No. (%) (n = 148)
No ILD	54 (36.5)
ILD CTCAE Grade 1 2 3	44 (29.7) 38 (25.7) 12 (8.1)
Withdrawal or discontinuation of durvalumab + -	54 (36.5) 40 (27.0)
Use of systemic corticosteroids + -	40 (27.0) 54 (36.5)

ILD, interstitial lung disease; CTCAE, Common Terminology Criteria for Adverse Events, version 5.0.

**Table 3 cancers-14-06236-t003:** Comparison of characteristics between patients with and without grade ≥2 ILD.

Patient Characteristics	ILD Grade ≤1, n = 98	ILD Grade ≥2, n = 50	*p*-Value
Age, years median (range)	71 (43–86)	73 (51–85)	0.003
Sex, (%) Male/female	67 (68.4)/31 (31.6)	39 (26.3)/11 (7.4)	0.219
BMI median (range)	20.9 (13.36–31.24)	21.37 (16.36–28.25)	0.150
ECOG-PS, (%) 0/≥1	77 (78.6)/21 (21.4)	34 (68.0)/16 (32.0)	0.160
Smoking status, (%) Non-smokers/former or current smokers	83 (56.1)/15 (10.1)	42 (84.0)/8 (16.0)	0.912
Brinkman Index median (range)	820 (0–2700)	920 (0–3840)	0.603
Autoimmune disease, (%) +/−	4 (4.1)/94 (95.9)	3 (6.0)/47 (94.0)	0.841
Histology, (%) Ad/non-Ad	55 (56.1)/43 (43.9)	22 (44.0)/28 (56.0)	0.163
Clinical stage, (%) II/III/IV/post operative recurrence	8 (8.2)/87 (88.8)/ 2 (2.0)/1 (1.0)	4 (8.0)/45 (90.0)/ 0 (0.0)/1 (2.0)	0.915
Driver mutation, (%) +/−/not tested	9 (9.2)/61 (62.2)/ 28 (28.6)	6 (12.0)/24 (48.0)/ 20 (40.0)	0.359
PD-L1 TPS, (%) <1%/1–49%/≥50%/not tested	18 (18.4)/23 (23.4)/ 31 (31.6)/26 (26.5)	11 (22.0)/12 (24.0)/ 10 (20.0)/17 (34.0)	0.440
Laboratory findings			
WBC count (/μL), median (range)	4100 (1700–8880)	3500 (1300–7000)	0.018
Lymphocyte count (/μL), median (range)	760 (210–2500)	675 (180–5000)	0.296
LDH level (IU/L), median (range)	194 (126–343)	187.5 (131–413)	0.942
CRP level (mg/dL), median (range)	0.52 (0–13.58)	0.475 (0–4.7)	0.440
KL-6 level (U/mL), median (range)	272 (116–991)	315 (135–2221)	0.067
Information of CRT			
Concurrent chemotherapy			0.472
Weekly carboplatin + paclitaxel	55 (56.1)	32 (64.0)	
Cisplatin + S-1	12 (12.2)	6 (12.0)	
Cisplatin + vinorelbine	24 (24.5)	10 (20.0)	
Daily carboplatin	4 (4.1)	1 (2.0)	
Carboplatin + paclitaxel	0 (0.0)	1 (2.0)	
Cisplatin + docetaxel	3 (3.1)	0 (0.0)	
Total irradiation dose, Gy median (range)	65 (40–70)	63 (50–70)	0.699
V5, %, (%) median (range)	40.4 (10.7–83.2)	47.6 (24–78.9)	0.001
V20, %, (%) median (range)	20.6 (2.9–40.7)	23.75 (7–32.3)	0.007
Mean lung dose, Gy median (range)	11.3 (2.8–19)	13.5 (5.3–19.2)	<0.001
Radiation technique 3D/IMRT/3D + IMRT	56 (57.1)/34 (34.7) /8 (8.2)	28 (56.0)/19 (38.0) /3 (6.0)	0.853
Best tumor response CR/PR/SD/PD/not evaluated	6 (6.1)/66 (67.3)/ 24 (24.5)/1 (1.0)/ 1 (1.0)	0 (0.0)/36 (72.0)/ 14 (28.0)/0 (0.0)/ 0 (0.0)	0.361
Interval between CRT and durvalumab, days median (range)	18.5 (1–54)	15 (1–54)	0.441
CT findings			
IP, (%) +/−	1 (1.0)/97 (99.0)	1 (2.0)/49 (98.0)	0.625
Emphysema, (%) +/−	50 (51.0)/48 (49.0)	32 (64.0)/18 (36.0)	0.133
ILAs, (%) +/−	26 (26.5)/72 (73.5)	30 (60.0)/20 (40.0)	<0.001
Type of ILAs			
GGA +/−	13 (13.3)/85 (86.7)	28 (56.0)/22 (44.0)	<0.001
Reticulation +/−	14 (14.3)/84 (85.7)	16 (32.0)/34 (68.0)	0.011
Honeycomb +/−	0 (0.0)/98 (100.0)	1 (2.0)/49 (98.0)	0.160
Other +/−	6 (6.1)/92 (93.9)	0 (0.0)/50 (100.0)	0.074

BMI, body mass index; ECOG PS, Eastern Cooperative Oncology Group performance status; Ad, adenocarcinoma; PD-L1, programmed death-ligand 1; TPS, tumor proportion score; WBC, white blood cell; LDH, lactate dehydrogenase; CRP, C-reactive protein; CRT, chemoradiotherapy; V5, the volume of lung parenchyma that received ≥5 Gy; V20, volume of lung parenchyma that received ≥20 Gy; CR, complete response; PR, partial response; SD, stable disease; PD, progressive disease; CT, computed tomography; IP, interstitial pneumonitis; ILA, interstitial lung abnormalities; GGA, ground glass attenuation.

**Table 4 cancers-14-06236-t004:** Univariate logistic regression analysis of risk factors for grade ≥2 ILD.

	Odds Ratio	95% Confidence Interval	*p*-Value
Age, years ≥65	4.36	1.58–12.05	0.005
ECOG-PS ≥1	1.726	0.80–3.71	0.162
WBC count (/µL) ≥3600 (cutoff)	0.536	0.27–1.07	0.077
KL-6 level (U/mL) ≥500	0.94	0.35–2.51	0.905
V5, % ≥43.1 (cutoff)	3.68	1.78–7.63	<0.001
V20, % ≥22.4 (cutoff)	2.81	1.39–5.68	0.005
MLD, Gy ≥12.3 (cutoff)	3.66	1.78–7.54	<0.001
Emphysema +	1.70	0.85–3.44	0.135
ILAs +	4.15	2.02–8.55	<0.001
Type of ILAs			
GGA +	8.31	3.71–18.66	<0.001
Reticulation +	2.82	1.25–6.49	0.013

ECOG PS, Eastern Cooperative Oncology Group performance status; WBC, white blood cell; V5, the volume of lung parenchyma that received ≥5 Gy; V20, volume of lung parenchyma that received ≥20 Gy; MLD, mean lung dose; ILA, interstitial lung abnormalities; GGA, ground glass attenuation.

**Table 5 cancers-14-06236-t005:** Multivariate logistic regression analysis of risk factors for grade ≥2 ILD.

Model A			
	Odds Ratio	95% Confidence Interval	*p*-Value
Age, years ≥65	2.90	0.99–8.58	0.053
V20, % ≥22.4	2.64	1.24–5.62	0.012
ILAs +	3.70	1.69–7.72	0.001
Model B			
	Odds Ratio	95% Confidence Interval	*p*-Value
Age, years ≥65	2.75	0.89–8.52	0.080
V20, % ≥22.4	2.68	1.20–6.00	0.016
GGA +	6.71	2.80–16.08	<0.001
Reticulation +	1.47	0.54–3.95	0.441

ILA, interstitial lung abnormalities; V20, volume of lung parenchyma that received ≥20 Gy; GGA, ground glass attenuation.

**Table 6 cancers-14-06236-t006:** Univariate logistic regression analysis of risk factors for early onset grade ≥ 2 ILD.

	Odds Ratio	95% Confidence Interval	*p*-Value
ILAs +	6.25	2.29–17.03	<0.001
V5, % ≥43.1	4.58	1.60–13.11	0.005
V20, % ≥22.4	3.32	1.27–8.64	0.014
MLD, Gy ≥12.3	2.38	0.94–6.04	0.066

ILD, interstitial lung disease; ILAs, interstitial lung abnormalities; V5, the volume of lung parenchyma that received ≥5 Gy; V20, volume of lung parenchyma that received ≥20 Gy; MLD, mean lung dose.

**Table 7 cancers-14-06236-t007:** Multivariate logistic regression analysis of risk factors for early onset grade ≥2 ILD.

	Odds Ratio	95% Confidence Interval	*p*-Value
ILAs +	6.19	2.22–17.24	<0.001
V20, % ≥22.4	3.27	1.20–8.92	0.021

ILD, interstitial lung disease; ILAs, interstitial lung abnormalities; V20, volume of lung parenchyma that received ≥20 Gy.

## Data Availability

The data presented in this study are available to interested researchers upon reasonable requests to the corresponding author based on ethical approval.

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
