# Peer review of "Pre-Existing Interstitial Lung Abnormalities Are Independent Risk Factors for Interstitial Lung Disease during Durvalumab Treatment after Chemoradiotherapy in Patients with Locally Advanced Non-Small-Cell Lung Cancer"

_cancers, 2022, doi:10.3390/cancers14246236_

Round 1

Reviewer 1 Report

The authors reviewed 148 patients from 10 centers and identified ILA was the significant risk factor of ILD for locally advanced NSCLCs who received Durvalumab after CRT. Overall, the writing quality is good and the statistical method is basically correct. However, I still have several concerns about this study.

1. What is the different between ILD and checkpoint inhibitor-related pneumonitis which are reported as an irAE in many RCTs.

2. The authors investigated the presence of ILA after CRT. How do you differentiate whether ILD is due to CRT or ICI?

3. For clarity to all readers, I suggest authors show all kinds of ILA instead of only the 3 kinds of ILA.

4. The previous study of the author's team has confirmed the predictive value of ILA in ILD in advanced NSCLC who received immunotherapy. This study only changed the population, its innovation and clinical value are general. I suggest to establish a predict model to identify high-risk population of early-onset ILD.  Providing advice on when high-risk populations should have CT examination, and intervene as early as possible to avoid grade 3-5 adverse reactions.

5. What is the follow-up plan? How often should patients have CT reexamination? The long interval time of 2 CT examinations may affect the severity of ILD.

Reviewer 2 Report

Dear authors,

Introduction and discussion section should be improved because these sections do not contain enough data.

Please add some relevant information.

Author Response

Reviewer 2

Dear authors,

Introduction and discussion section should be improved because these sections do not contain enough data.

Please add some relevant information.

Response:

Thank you for your advice. We added the reference No.22 to introduction and discussion sections.

Reviewer 3 Report

Journal of Cancers

Research Article;

The article entitled “Pre-Existing Interstitial Lung Abnormalities are Independent Risk Factors for Interstitial Lung Disease during Durvalumab Treatment after Chemoradiotherapy in Patients with Locally Advanced Non-Small-Cell Lung Cancer’’. The authors investigate the The standard treatment for locally advanced non-small-cell lung cancer followed treatment with durvalumab. As interstitial lung disease is a life-threatening toxicity caused by CRT and durvalumab. Interstitial lung abnormalities manifest as minor interstitial shadows on computed tomography. the patients with non-small-cell lung cancer who received durvalumab after CRT. The patient data gives wonderful analysis. The prevalence of ILAs before durvalumab treatment was 37.8%. Among 148 patients, 63.5% developed ILD during durvalumab therapy. Univariate logistic regression analysis revealed that older age, high dose-volume histogram parameters, and the presence of ILAs were significant risk factors for grade 2 or higher ILD. Multivariate analysis showed that ILAs were independent risk factors for grade 2 or higher ILD. Pre-existing ILAs are risk factors for ILD during durvalumab treatment after CRT.

I carefully read the manuscript and found it suitable for publication in the journal. I accept this article for possible publication. The article could be considered for publication in the prestigious Cancers Journal.

Comments for Authors

Ø  The author needs to include the supplementary figure in the manuscript

Ø  The author didn’t determine that during the three years, could the patients receive any other self-treatment.

Ø  The impact of food and climate on the treatment efficacy?

Ø  It could be better to discuss the effect of side drugs, i-e antibiotics on the Durvalumab during treatment

Ø  The need to show the importance of the study in conclusion

Cite the following references.

v  https://doi.org/10.1038/s41419-021-03771-z
